# Role of Mitochondria in the Regulation of Effector Functions of Granulocytes

**DOI:** 10.3390/cells12182210

**Published:** 2023-09-05

**Authors:** Nina V. Vorobjeva, Maria A. Chelombitko, Galina F. Sud’ina, Roman A. Zinovkin, Boris V. Chernyak

**Affiliations:** 1Department Immunology, Biology Faculty, Lomonosov Moscow State University, 119234 Moscow, Russia; nvvorobjeva@mail.ru; 2Belozersky Institute of Physico-Chemical Biology, Lomonosov Moscow State University, 119992 Moscow, Russia; chelombitko@mail.bio.msu.ru (M.A.C.); roman.zinovkin@gmail.com (R.A.Z.); 3The Russian Clinical Research Center for Gerontology, Ministry of Healthcare of the Russian Federation, Pirogov Russian National Research Medical University, 129226 Moscow, Russia

**Keywords:** granulocytes, degranulation, oxidative burst, apoptosis, extracellular traps, leukotrienes, inflammation, mitochondria, mitochondrial ROS production, mitochondria-targeted antioxidants

## Abstract

Granulocytes (neutrophils, eosinophils, and basophils) are the most abundant circulating cells in the innate immune system. Circulating granulocytes, primarily neutrophils, can cross the endothelial barrier and activate various effector mechanisms to combat invasive pathogens. Eosinophils and basophils also play an important role in allergic reactions and antiparasitic defense. Granulocytes also regulate the immune response, wound healing, and tissue repair by releasing of various cytokines and lipid mediators. The effector mechanisms of granulocytes include the production of reactive oxygen species (ROS), degranulation, phagocytosis, and the formation of DNA-containing extracellular traps. Although all granulocytes are primarily glycolytic and have only a small number of mitochondria, a growing body of evidence suggests that mitochondria are involved in all effector functions as well as in the production of cytokines and lipid mediators and in apoptosis. It has been shown that the production of mitochondrial ROS controls signaling pathways that mediate the activation of granulocytes by various stimuli. In this review, we will briefly discuss the data on the role of mitochondria in the regulation of effector and other functions of granulocytes.

## 1. Introduction

Granulocytes (neutrophils, eosinophils, and basophils) are the most abundant circulating cells in the innate immune system. These myeloid polymorphonuclear cells are characterized by the presence of specific cytoplasmic granules that can be expelled into the environment upon activation. Circulating granulocytes can cross the endothelial barrier into various tissues and activate numerous effector mechanisms to combat invasive inflammatory pathogens. Eosinophils and basophils also play an important role in allergic (hypersensitive) reactions and antiparasitic defense. Another function of granulocytes is to regulate the immune response, wound healing, and tissue repair by releasing various cytokines and lipid mediators (prostaglandins, leukotrienes, resolvins, platelet activating factor, etc.). The effector mechanisms of granulocytes include the production of reactive oxygen species (ROS), degranulation (resulting in the release of lytic and prooxidant enzymes, as well as histamine), phagocytosis, and the formation of extracellular traps (ETs).

Granulocytes are mainly glycolytic, depending mainly on glycolysis and not oxidative phosphorylation for energy production [1,2,3,4]. Dependence on glycolysis is most pronounced in neutrophils, whereas eosinophils show high metabolic flexibility and use oxidative phosphorylation for effector functions [4]. However, even in neutrophils, mitochondrial ATP production may be important for antimicrobial protection. For example, it has been reported [5] that in *N*-formylmethionyl-leucyl-phenylalanine (fMLP)-stimulated neutrophils, mitochondrial-produced ATP activates early autocrine purinergic signaling that supports respiratory burst, degranulation, and phagocytosis. Data on the metabolic phenotype of basophils are limited. Sumbayev and colleagues [6] demonstrated that IgE-mediated activation of primary human basophils is accompanied by the accumulation of HIF-1a, which is known to regulate glycolysis. Immune metabolism has been studied in the RBL-2H3 cell line, which shares features of mast cells and basophils. The results of these studies revealed the need for both glycolysis and oxidative phosphorylation in IgE-dependent activation of RBL-2H3 cells [7,8].

All granulocytes have only a small number of mitochondria, and neutrophils have fewer mitochondria than the other cell types. The functionality of mitochondria in granulocytes has long been questioned because the respiration rate in these cells is low and difficult to measure. The mitochondrial membrane potential (ΔΨ), which is an indicator of their functional state, has also not been correctly measured due to the high membrane potential at the plasma membrane. During activation, granulocytes are characterized by strong changes in the potential of the plasma membrane (positive inside) due to the electrogenic activity of NADPH oxidase and the opening of ion channels. The first reliable measurements of ΔΨ became possible after the invention of mitochondrial fluorescent voltage-sensitive probes [9]. In neutrophils, an unusual ΔΨ generation mechanism has been described based on the activity of mitochondrial glycerol-3-phosphate dehydrogenase, which reduces coenzyme Q and stimulates electrogenic electron flow through complexes III and IV of the electron transport chain [10]. It is important to emphasize that the mitochondrial membrane potential is of decisive importance not only for the formation of ATP but also for the transport of Ca^2+^ into mitochondria and for the reduction of NAD^+^ by complex I (reverse electron transfer), which is accompanied by the formation of ROS. Mitochondria in granulocytes can form dynamic networks, and a decrease in ΔΨ leads to the fragmentation of elongated organelles. It is assumed that mitochondrial dynamics plays an important role in the regulation of the functions of neutrophils [11], eosinophils [12], basophils [8], and mast cells [13].

The production of reactive oxygen species (ROS) in mitochondria is known as an important regulator of many processes in various cell types. It is catalyzed by redox sites in complex I (NADH:ubiquinone oxidoreductase) and complex III (ubiquinone:cytochrome c oxidoreductase), as well as by several dehydrogenases in the mitochondrial matrix (pyruvate dehydrogenase, 2-oxoglutarate dehydrogenase [14]. In granulocytes, the main source of ROS production is the nonmitochondrial enzyme NADPH oxidase (NOX2). NADPH oxidase is silent in resting granulocytes and is rapidly activated upon stimulation due to phosphorylation by several protein kinases, such as protein kinase C (PKC). Activation of NADPH oxidase, together with degranulation in neutrophils and eosinophils, leads to a massive release of ROS and a strong increase in oxygen consumption (respiratory burst) [15]. Human basophils do not develop respiratory burst in response to various stimuli and do not express NADPH oxidase [16,17]. A much less intense production of mitochondrial ROS (mtROS) may be an important signaling event initiating the activation of NADPH oxidase. The role of mtROS in NOX2 activation, degranulation, and other effector functions of neutrophils and basophils will be discussed in detail in this review. The possible role of mitochondria in NOX2 activation in eosinophils, to the best of our knowledge, has not yet been studied. This task is interesting, since it is known that during a respiratory burst, eosinophils produce two to three times more ROS than neutrophils [18]. At least in part, this difference reflects a more sustained respiratory burst in eosinophils. It has been shown that eosinophils preferentially assemble and activate NOX2 at the plasma membrane, while neutrophils significantly activate it in intracellular vesicles [19]. This difference may be due to the different role of these cells in defense against pathogens: Whereas neutrophils actively phagocytize microbial pathogens and kill them in phagosomes with the help of ROS, eosinophils preferentially release ROS to the outside in the fight against parasites.

The production of cytokines and lipid mediators is not included in the effector functions of granulocytes but is very important in their physiology. In particular, a wide variety of cytokines secreted by neutrophils are involved not only in inflammation but also in immunomodulation, hematopoiesis, angiogenesis, and regeneration [20]. Immunosuppressive neutrophils (largely coinciding with granulocytic/polymorphonuclear myeloid-derived suppressor cells) are of great interest because they may stimulate tumor growth and interfere with cancer immunotherapy [21,22]. Immunosuppression by neutrophils, mediated in part by cytokines, mainly depends on their ability to inhibit T cell proliferation and activation, as well as stimulation of regulatory T cells. Importantly, inhibition of mitochondrial functions has been reported to selectively inhibit neutrophil immunosuppressive functions and stimulate T cell response [23]. Cytokines produced by eosinophils are involved in allergen-induced inflammation and in the promotion of type 2 immune responses [24]. Basophils are the main source of the cytokine IL-4, which plays a crucial role in the regulation of Th2-immune response and allergic inflammation. In addition to IL-4, basophilic granulocytes secrete cytokines such as IL-13, IL-6, TNFα, and TSLP, as well as a number of chemokines [25]. The expression of most cytokines depends on the activity of the transcription factor NF-kB. It was found that in non-immune cells, NF-kB activation depends on the production of mtROS [26], but this possibility has not been studied in granulocytes.

Among the lipid mediators (eicosanoids) produced by granulocytes, the most important role belongs to leukotrienes, which are produced by lipoxygenases that catalyze the insertion of oxygen at positions 5 and 15 of polyunsaturated fatty acids. Leukotriene B4 (LTB4), which is produced by the 5-lipoxygenase (5-LOX) pathway in all granulocytes, regulates neutrophils’ collective behavior as well as various aspects of inflammation (see below). Cysteine-containing leukotrienes, produced by 5-LOX in eosinophils, basophils, and mast cells, are potent bronchoconstrictors and vascular leak stimulators. Eosinophils also express 15-LOX and produce lipoxins and resolvins, which promote the resolution of inflammatory processes [27,28]. The possible role of mitochondria in the production of leukotrienes has not been studied until recently. Our studies using SkQ1 show that mtROS are critical for 5-LOX activation and LTB4 production in neutrophils [29].

All granulocytes are short-lived cells, and the exact values of their half-life in the bloodstream are a matter of controversy even for the most studied of them: neutrophils [30]. Estimates range from 7–9 h to 3.75–5 days. Eosinophils in the blood also have a short half-life of 8 to 18 h, but in tissues it increases to at least 5 days. Moreover, their in vitro lifespan can be extended to 14 days or more with the help of cytokines [31]. The lifespan of basophils in the blood is estimated to be approx. 60 h [32]. Spontaneous apoptosis is critical to the normal physiology of granulocytes, and mitochondria play a central role in orchestrating apoptosis signaling, mainly through the release of several pro-apoptotic proteins into the cytoplasm. It is important to note that granulocyte apoptosis depends on the ROS produced by them [33] or by other cells [34]. This review will discuss recent studies pointing to a role for mtROS in granulocyte apoptosis.

The mitochondrial control of granulocyte effector functions, leukotriene synthesis, and apoptosis is illustrated by the scheme (Figure 1).

## 2. Degranulation

Degranulation is one of the key effector functions of granulocytes and is defined as the exocytosis of pre-formed substances from granules. Although different granulocytes (neutrophils, eosinophils, basophils) have generally similar mechanisms of degranulation, there are some peculiarities in their exocytosis mechanisms [35].

In neutrophils, the mechanisms of degranulation are best understood. These cells contain at least four distinct types of granules, which are (1) primary or azurophilic granules, which contain elastase, myeloperoxidase (MPO), cathepsins, and defensins; (2) secondary or specific granules, which include lactoferrin and components of NADPH oxidase—gp91phox and p22phox; and (3) tertiary or gelatinase granules, which store matrix metalloproteinases (MMPs) [36]. More recently, a fourth granule type has been described, which is enriched in the microbial lectin ficolin-1 [37]. Neutrophils also contain secretory vesicles, which are formed by endocytosis.

In eosinophils, two different types of granules have been described in the cytoplasm to date [38,39]. Secondary granules, also termed specific or crystalloid granules, consist of a crystalloid core and matrix and include major basic protein, eosinophil cationic protein, eosinophil peroxidase (EPO)—the analog of MPO, and eosinophil-derived neurotoxin. Larger or primary granules are coreless and include crystalline Charcot–Leiden protein or galectin 10 and EPO.

Basophils contain dense cytoplasmic granules, which include a broad array of allergic inflammation mediators such as histamine, chondroitin sulfate, and basogranulin [40].

During receptor-mediated stimulation of neutrophils, the granules are mobilized, causing secretion of their contents, which can occur intracellularly towards phagosomes or extracellularly. Three different mechanisms of granule exocytosis have been described in human neutrophils [41]. These include (1) the fusion of granules independently with the plasma membrane, which is called simple exocytosis; (2) granules fusing with each other in the cytosol, followed by their interaction with the plasma membrane, which is called compound exocytosis; and (3) the initial fusion of granules with the plasma membrane, followed by fusion of subsequent granules with the membrane of the previous one, resulting in the formation of a degranulation sac. This type of exocytosis has been termed cumulative fusion [41].

Eosinophils and basophils possess, in addition to the aforementioned types of exocytosis found in neutrophils, a piecemeal degranulation, which is characterized by the release of secondary granules without the fusion of vesicles and plasma membranes [42]. In addition, all granulocytes can release granules after cytolysis (various forms of necrotic death, including the release of extracellular traps (ETosis)).

The content of neutrophil granules, such as MPO, elastase, lactoferrin, and MMP, in addition to microbicidal activity, is also highly cytotoxic. Therefore, to ensure that the content of the various granules is released at the right time and in the right place, exocytosis needs to be finely regulated in order to avoid damage to surrounding tissues. This regulation occurs at several stages of degranulation, with the participation of various mediators. The most powerful neutrophil activators include *N*-formyl peptides of bacterial or mitochondrial origin. These peptides interact with G protein-coupled receptors (GPCRs) on the surface of neutrophils, followed by stimulation of phospholipase C (PLC), phospholipase D (PLD), and phospholipase A2 (PLA2). PLC then cleaves phosphatidylinositol 4,5-bisphosphate (PIP2) with the formation of inositol trisphosphate (IP3) and diacylglycerol, which induce the release of Ca^2+^ from the intracellular stores and the activation of conventional protein kinase (PKC) isoforms, respectively. Activation of PLD leads to the production of phosphatidic acid, which can activate several protein kinases, such as PKC and mTOR. Activation of PLA2 results in arachidonic acid production, which is involved in the synthesis of leukotrienes and prostaglandins, stimulates PKC, or directly stimulates NADPH oxidase [43]. An increase in intracellular Ca^2+^ is a powerful inducer of granule exocytosis in neutrophils [44].

Another signaling pathway of GPCR-mediated activation is Ca^2+^ independent. It triggers the activation of Src-family tyrosine kinases (Hck, Fgr, and Lyn), where Hck and Fgr are involved in granule traffic to the target membranes. In addition, both kinases have been shown to regulate the activation of guanine nucleotide exchange factor of small GTPases, Vav 1, which is involved in the activation of Rac1 and Rac2 and also induces actin polymerization and oxidative burst [45].

The GPCR-mediated pathway also activates PI3-kinase-γ (PI3Kγ), which is engaged in the production of PIP2, a mediator of degranulation. Thus, both Ca^2+^-dependent and Ca^2+^-independent signaling pathways coordinate the activation of neutrophil degranulation [46].

In addition to above-mentioned mediators of degranulation, mitogen-activated protein kinase (MAPK) pathways (including extracellular signal-regulated kinase (ERK), c-Jun *N*-terminal kinase (JNK), and p38 MAPK) have been shown to play a pivotal role in granulocyte degranulation induced by various stimuli [47,48].

We hypothesized that mtROS may be an additional regulator of receptor-mediated degranulation of human neutrophils. The prerequisite for this hypothesis was the study by Fossati and co-workers [49], who reported that the dissipation of mitochondrial membrane potential with an uncoupler of oxidative phosphorylation leads to a decline in the oxidative burst, phagocytosis, and degranulation. To test our hypothesis, we used the mitochondria-targeted antioxidant SkQ1, which accumulates in mitochondria due to the membrane potential [50]. Using its fluorescent analog, SkQR1, we showed that this antioxidant selectively accumulates in the mitochondria of neutrophils [51]. It was shown that SkQ1 effectively inhibited fMLP-dependent mtROS production measured with mitochondria-targeted superoxide-sensitive fluorescent dye MitoSOX in the presence of a catalase that was added to remove extracellular hydrogen peroxide. Finally, SkQ1 inhibited fMLP-induced exocytosis of azurophil and specific granules, whereas its analog without an antioxidant quinol residue, C_12_TPP, was ineffective [51]. These results indicate that the central events of GPCR-mediated activation of human neutrophils, such as degranulation, are dependent on mtROS production.

The degranulation of basophils can be induced by binding IgE/antigen complexes to high-affinity IgE receptors FcεRI. It is mediated by the activation of the Src-family kinases (Fyn, Lyn, and Syk), PI3-kinase, and MAPK-dependent pathways [25]. It has been shown that IgE-dependent degranulation of human peripheral blood basophils is accompanied by the formation of superoxides [52,53], but the possible role of ROS in degranulation has not been established.

Since basophils make up less than 1% of peripheral blood leukocytes, studying them is difficult. Several cell lines have been developed that are considered useful models of basophils and mast cells. Thus, the cell line of basophilic leukemia RBL-2H3 expresses a high-affinity IgE receptor and is widely used to study the mechanisms of IgE-dependent degranulation [54]. It has been shown that IgE-dependent degranulation of RBL-2H3 cells is accompanied by the formation of superoxide [55] in the same way as in basophils; however, unlike basophils, RBL-2H3 cells express NADPH oxidase like mast cells. Diphenylene iodonium (DPI), a non-selective inhibitor of flavin-containing enzymes that inhibits both NADPH oxidase and mitochondrial sources of ROS, reduces IgE-dependent degranulation of RBL-2H3 cells [55]. The production of mitochondrial ROS during the activation of RBL-2H3 cells has also been shown in several studies [8,56]. As mentioned above, basophils do not express NADPH oxidase [16,17]; thus, it can be assumed that mitochondria are the main source of ROS involved in the regulation of basophil degranulation.

SkQ1 and its analogue without the antioxidant moiety C_12_TPP at very low concentrations inhibit IgE-dependent degranulation of RBL-2H3 cells and prevent mitochondrial dysfunction, including fragmentation and a drop in ATP level [8]. Since fragmentation of mitochondria is associated with an increase in the production of mtROS [57] it is possible that SkQ1 and C_12_TPP may prevent mitochondrial dysfunction by reducing the mtROS level. This antioxidant effect is probably mediated by a partial (mild) uncoupling caused by TPP-based compounds [58]. These data indicate that an increase in the level of mtROS in RBL-2H3 cells is necessary for IgE-dependent degranulation. At the same time, an increase in the level of mtROS by mitochondria-targeted curcuminoids Mitocur-1 and Mitocur-3 does not lead to spontaneous degranulation and does not enhance IgE-dependent degranulation of RBL-2H3 cells [59]. These data indicate that the relationship between mtROS and basophil degranulation needs further study.

Various signaling pathways involved in the stimulation of the degranulation of granulocytes are redox sensitive and could potentially be the targets of mtROS. Thus, PKC has two redox-sensitive sulfur-zinc clusters in the diacylglycerol-binding domain [60]. Some members of Src-family kinases (in particular, Lyn kinase) are activated by ROS [61]. PI3K- and MAPK-dependent pathways also include redox-sensitive components, but the primary target of mtROS action remains unknown.

## 3. Oxidative Burst

Receptor-mediated stimulation of neutrophils and eosinophils promotes not only granule exocytosis pathways but also the assembly and activation of NOX2. NOX2 is a multisubunit enzymatic complex that becomes active after the association of four cytosolic subunits—p47phox, p67phox, p40phox, and Rac2—with transmembrane proteins gp91phox and p22phox (cytochrome b_557_). The assembled and activated NADPH oxidase is responsible for the formation of superoxide (O_2_^–^), which is converted to hydrogen peroxide. H_2_O_2_ can be transformed into hypochlorous acid by MPO. All of these compounds belong to ROS, and the abrupt formation of ROS and the corresponding increase in oxygen uptake is called an oxidative or respiratory burst.

The assembly and activation of NADPH oxidase in neutrophils and eosinophils are very similar. However, human eosinophils have been shown to produce all ROS extracellularly, whereas neutrophils predominantly produce ROS intracellularly during phagocytosis [19].

The activation of NADPH oxidase by mtROS was demonstrated for the first time by Dikalov and colleagues [62] on endothelial cells. It was shown both in cell culture and in vivo that the activation of NADPH oxidase induced with hormone angiotensin II was inhibited by the mitochondria-targeted antioxidant mitoTEMPO and by the overexpression of the mitochondrial superoxide dismutase MnSOD (SOD2). Later, it was demonstrated that NOX2 is the only isoform stimulated by mtROS in the endothelium [63]. As for the activation of NOX2 in human neutrophils, it was found that exogenous hydrogen peroxide induced a dose-dependent stimulation of superoxide production and accelerated an oxidative burst in response to phorbol 12-myristate 13-acetate (PMA), indicating that a feedback loop is involved in neutrophil activation [64]. In addition, studies of PMA-induced activation of human immortalized lymphoblast cell lines have shown that mtROS are involved in the stimulation of NADPH oxidase [65]. Additional evidence was presented by Daiber and colleagues [66], who described the activation of NOX2 in human neutrophils by mtROS, induced by myxothiazol, the inhibitor of respiratory chain complex III. It was also shown that translocation of p47phox, p67phox, and Rac1 subunits of NOX2 to the membrane, as well as NADPH oxidase activation in response to myxothiazol, was inhibited by the mitochondria-targeted antioxidant mitoTEMPO [66]. Angiotensin II-dependent NOX2 activation in the endothelium and myxothiazol-dependent activation in neutrophils [66] were prevented by specific inhibitors of the mitochondrial permeability transition pore (mPTP). The opening of this large non-selective proteinaceous pore in the inner mitochondrial membrane can be caused by Ca^2+^ overload and some stresses, leading to various cellular dysfunctions [67]. Mitochondrial cyclophilin D (CypD) acts as a Ca^2+^ sensitizer for mPTP opening, and its knockout in mice prevents the development of angiotensin II-dependent hypertension [66,68] and myxothiazol-dependent oxidative burst in whole blood [66]. Studies on endothelial cells show that angiotensin II-dependent mtROS production depends on the opening of mPTP [68], whereas the role of mPTP in neutrophil activation remains unclear.

We have analyzed the possible role of mtROS in receptor-stimulated activation of NOX2 in neutrophils using the mitochondria-targeted antioxidant SkQ1 [51,69]. It was shown that total ROS production during fMLP-induced oxidative burst measured using luminol-enhanced chemiluminescence was prevented with SkQ1, whereas C_12_TPP had no inhibitory effect [51]. Using a pharmacological inhibitor approach, we demonstrated the critical role of the mPTP in receptor-stimulated mtROS production and the oxidative burst of human neutrophils [69]. This conclusion was based on the application of several specific inhibitors of the mPTP opening with different mechanisms of action, such as sangliferin A (a CypD inhibitor) and bongkrekic acid (an inhibitor of ATP/ADP transporter involved in mPTP function). These inhibitors suppressed the production of mtROS and the oxidative burst induced not only with fMLP but also with calcium ionophore A23187 [69], so we hypothesize that the increase in cytoplasmic Ca^2+^ induced by A23187 and fMLP caused the opening of mPTP. Excess formation of mtROS is probably mediated by the release from mitochondria of the main components of antioxidant defense, such as NADPH [70] and reduced glutathione [69]. Along with Ca^2+^, oxidative stress is an effective inducer of the mPTP opening [67]. It is possible that ROS formed by NADPH oxidase may penetrate the cell and stimulate mPTP opening, thus creating an amplification loop leading to the activation of NADPH oxidase.

Overall, our results indicate that mtROS are critical for receptor-dependent activation of NADPH oxidase. However, the primary targets of mtROS remain unknown. The oxidative burst in granulocytes may be mediated by the same redox-sensitive pathways (PKC, Src, MAPK) as degranulation. In addition, an important component of NADPH oxidase, p47phox, has been shown to be redox sensitive. ROS-dependent association of protein disulfide isomerase with p47phox leads to its translocation to the membrane with subsequent assembly of NADPH oxidase [71].

The generation of large amounts of ROS by NOX2 upon granulocyte activation (oxidative burst) is an important weapon for killing various pathogens. This function can be exemplified by the consequences of chronic granulomatous disease (CGD), where deficiency of subunits of the NOX2 complex leads to severe and chronic infections in patients [72]. At the same time, ROS generated by intracellular NOX2 or formed from the NOX2-derived extracellular hydrogen peroxide that has penetrated into the cell can strongly affect various redox-sensitive signaling pathways and functions of granulocytes. In particular, it has recently been found that neutrophils can influence adaptive immunity by participating in the functioning of the lymph nodes [73]. Neutrophils can influence antigen presentation indirectly by interacting with dendritic cells and macrophages or directly by expressing the major histocompatibility complex for antigen presentation. ROS production by NOX2 in antigen-presenting cells has been shown to promote antigen presentation [74]. Presumably, impairment of this NOX2 function in CGD patients leads to the commonly observed symptoms of autoimmunity in addition to impairment of antipathogenic defenses.

## 4. The Role of Mitochondria in Extracellular Trap Formation

### 4.1. Neutrophil Extracellular Traps (NETs) and NETosis

Almost two decades ago, Arturo Zychlinsky and co-workers [75] described a new effector function of neutrophils, the release of neutrophil extracellular traps (NETs). NETs consist of decondensed chromatin decorated with antimicrobial proteases from granules, the cytoplasm, and the nucleus. The antimicrobial effect of NETs is due to the restriction of pathogens spreading throughout the organism or even their destruction in situ. Since it was initially shown that NET formation is accompanied by cell death, this process is called NETosis [76]. The formation of NETs, in addition to host defensive function, plays an essential role in the pathogenesis of autoimmune and inflammatory disorders, such as systemic lupus erythematosus, rheumatoid arthritis, small vessel vasculitis, and psoriasis [77,78,79,80,81]. NETs are also involved in thrombosis, a variety of pulmonary pathologies, chronic rhinosinusitis, sepsis, and malignancies [80,81,82].

The release of chromatin was also found in other types of granulocytes, including eosinophils [83] and basophils [17], as well as in mast cells [84], T and B lymphocytes [85], monocytes [86], and macrophages [87].

The formation of NETs can be activated by various physiological stimuli, such as bacteria, fungi, protozoa, viruses, and bacterial products (lipopolysaccharides (LPS)). NETs can be also induced by antibodies and immune complexes, cytokines and chemokines (IL-8, TNF-α, IFN-γ), complement components, etc.

Currently, two fundamentally different forms of NET formation are described: the classical NETosis, and vital NET release, in which cells retain their viability and other effector functions. The formation of vital NETs occurs through the budding of nuclei and the release of vesicles filled with DNA [88]. In addition to nuclear DNA-containing NETs, vital NETs can be formed due to the release of mitochondrial DNA [89,90].

Classic or suicidal NETosis is a multistep sequentially developing process that involves the dissociation of protein complex “azurosomes” located in the membranes of azurophilic granules and containing various enzymes, including neutrophil elastase (NE), cathepsin G, and myeloperoxidase (MPO) [91]. The release of NE and MPO into the cytoplasm and their subsequent migration into the nucleus, accompanied by activation of the histone-citrullinating enzyme peptidylarginine deiminase 4, promote the decondensation of nuclear chromatin [75]. These events culminate in the release of chromatin from the nucleus, rupture of the cell membrane, and NET release [75].

The signaling pathways leading to NETosis can include the activation of various isoforms of PKC [92,93,94], cyclin-dependent kinases 4/6 [95], Raf-MEK-ERK-signaling cascade [96], and Src-family tyrosine kinases [97], but the detailed picture remains unclear.

NETosis induced by various stimuli depends on ROS generated by NADPH oxidase [75]. However, some stimuli, including several microorganisms [98,99], monosodium urate crystals [100], activated platelets or platelet-derived microparticles [101], complement components [102], cytokines [103], and soluble immune complexes [104] induce NETosis independently of NOX2-derived ROS. NOX2-independent NETosis also can be induced by calcium ionophores A23187 and ionomycin [105]. Importantly, this form of NETosis still requires ROS, and these are mtROS [51,105,106]. In particular, in neutrophils isolated from the blood of patients with X-linked CGD lacking functional NOX2, the formation of NETs in response to A23187 was prevented by mitochondria-targeted antioxidant SkQ1 [69] and therefore depends on the enhanced generation of mtROS without the participation of NADPH oxidase. At the same time, NETosis induced by A23187 in neutrophils from healthy donors was suppressed by both SkQ1 and specific NADPH oxidase inhibitors such as apocynin and VAS2870 [69], indicating that mtROS can stimulate NETosis via the activation of NADPH oxidase. We assume that in NADPH oxidase-deficient neutrophils, mtROS are produced with increased intensity due to the excessive accumulation of Ca^2+^ [107] and that their amount is sufficient to trigger NETosis, whereas in normal cells the action of mtROS depends on the activation of NOX2. In both cases, mtROS generation responsible for NETosis was shown to be dependent on the mPTP opening [69].

As shown in our study [69], mtROS generation induced by both fMLP and A23187 depends on the opening of mPTP. This phenomenon remained unnoticed for a long time, probably due to the fact that the most well-known mPTP inhibitor, cyclosporin A, is also an inhibitor of cytoplasmic phosphatase calcineurin, which is involved in the transmission of numerous Ca^2+^-dependent signals [67]. In our study, we used the mPTP inhibitors sangliferin A and bongkrekic acid, which did not affect calcineurin. Interestingly, fMLP did not cause significant decrease in the mitochondrial membrane potential and mitochondrial swelling characteristic for the prolonged opening of mPTP. We assume that the short-term increase in cytoplasmic Ca^2+^ caused by fMLP induced temporal reversible pore opening. This mode of mPTP opening has been described for both isolated mitochondria [67] and cell models [70]. Ca^2+^ overload caused by A23187 in neutrophils presumably leads to long-term opening of the mPTP, as revealed by the significant swelling of mitochondria that precedes chromatin decondensation and destruction of the nuclear envelope [69]. The regulation of effector functions of neutrophils by the mPTP opening and mitochondrial ROS is illustrated by the scheme (Figure 2).

Recently, Dunham-Snary and co-workers [108] reported that functional mitochondria from healthy donor neutrophils are involved in the defense against *Staphylococcus aureus* infection. The authors showed that inhibition of electron transport chain complex III impairs infection-induced mtROS formation, NET formation (both vital and lytic), and bacterial killing. In addition, the authors described a surprising mechanism of bacterial engulfment involving mitochondria, which formed lasso-like structures around pathogens prior to phagocytosis [108].

### 4.2. The Mechanisms of Eosinophil and Basophil Extracellular Trap Formation

Eosinophils, like neutrophils, can release DNA-containing fibrils upon stimulation, which have been termed eosinophil extracellular traps (EETs). In eosinophils, two different mechanisms of EET formation have been described. One of them (EETosis) leads to cell death similar to NETosis and depends on ROS generated by NADPH oxidase [109]. Another mechanism for EET release has been studied in detail by Simon and colleagues [83,89] and is based on the rapid extrusion of mitochondrial DNA (mtDNA). This mechanism is independent of cell death and is preceded by degranulation [110]. Vital release of EETs also depends on the activation of NADPH oxidase as well as on PI3K-dependent signaling [111]. The possible role of mtROS in both mechanisms of ET formation was not studied.

It has recently been reported that the physiological activation of human and mouse basophils can also lead to the rapid formation of extracellular DNA fibrils [17], so-called basophil extracellular traps (BETs). It was shown that basophils release mtDNA rather than nuclear DNA, and this process is similar to vital NETosis and depends on ROS. Mitochondria-targeted antioxidant MitoQ inhibits BET formation, indicating that mtROS are critical [17]. Since NADPH oxidase is absent in basophils [16,17], mitochondria are probably the only source of ROS that controls the formation of BETs.

## 5. Leukotriene Synthesis in Granulocytes

The key enzyme in the formation of leukotrienes in granulocytes is 5-lipoxygenase (5-LOX), which catalyzes the insertion of oxygen at position 5 of arachidonic acid to form 5S-hydroperoxy-6E,8Z,11Z,14Z-eicosatetraenoic acid (5-HpETE) [112]. The 5-HpETE is further converted to the epoxide intermediate leukotriene A4 (LTA4) by 5-LOX with the aid of 5-LOX-activating protein (FLAP) [113]. The unstable LTA4 can be transformed further to the dihydroxy compound leukotriene B4 by LTA4 hydrolase or to glutathione-conjugated leukotriene C4 by LTC4 synthase. Leukotriene C4 can be metabolized to LTD4 by enzymatic elimination of glutamic acid, and LTD4 is hydrolyzed to LTE4 [113].

The main 5-LOX metabolite in neutrophils is leukotriene B4 [114]. It is the strongest chemoattractant for neutrophils and eosinophils, acting in concentrations of the order of 10^−14^ M [115]. LTB4 is secreted by neutrophils stimulated by primary chemoattractants such as fMLP, which enhance neutrophil chemotaxis [116]. LTB4-dependent chemotaxis is responsible for the coordinated behavior of neutrophils, called swarming, when cells gather around a microbial cluster or a large pathogen [117]. For example, the highly pathogenic yeast *Candida albicans* stimulates LTB4 synthesis and the swarming of neutrophils only during the formation of long filamentous hyphae [118]. We recently found that LTB4 synthesis in neutrophils in the presence of bacteria and fMLP induces the formation of intercellular contacts [119], which may be a prerequisite for further clustering and swarming. LTB4-dependent swarming of neutrophils is an important defense mechanism, but excessive swarming leads to severe pathologies. For example, it has been shown that pulmonary capillaritis in lethal fungal sepsis is suppressed in mice lacking the LTB4 receptor [120].

At higher concentrations, LTB4 also stimulates extravasation, phagocytosis, adhesion of circulating neutrophils to the vascular endothelium, degranulation, and aggregation of granulocytes [121]. LTB4 also enhances the activation, proliferation, and differentiation of B-lymphocytes and the activity of killer cells and induces the secretion of interleukin-1 by macrophages and the production of γ-interferon [122]. Leukotriene B4 is also produced by eosinophils and contributes to the pathogenesis of asthma [115,123]. Under conditions of oxidative stress, 5-HETE is transformed by 5-hydroxyeicosanoid dehydrogenase to 5-oxo-ETE (5-oxo-6,8,11,14-eicosatetraenoic acid), which is a potent eosinophil chemoattractant. Receptors for 5-oxo-ETE (OXE receptors) are expressed on all granulocytes, and antagonists of these receptors inhibit allergen-induced eosinophil infiltration [124]. Cysteine-containing leukotrienes (cysLTs) C4, D4, and E4 are synthesized mainly by eosinophils [125], basophils [126], and mast cells. CysLTs stimulate bronchoconstriction and vascular leakage as well as inflammatory responses, which greatly contribute to the development of asthma and other allergic diseases [127].

The induction of LTB4 synthesis depends on the translocation of 5-LOX to the nuclear membrane. This process depends on an increase in intracellular Ca^2+^ and on 5-LOX phosphorylation by p38 MAPK and ERK1/2 kinases [112]. We recently showed [29] that SkQ1 prevents the activation of 5-LOX by fMLP, A23187, or zymosan (protein-carbohydrate complexes of the yeast cell wall). This effect was mediated by mtROS scavenging and did not depend on changes in the cytoplasmic Ca^2+^ concentration. The effect of SkQ1 was not mediated by inhibition of NADPH oxidase. These results are fully consistent with earlier data on the role of ROS in 5-LOX activation [128] and with high production of LTB4 in mice lacking functional NOX2 [129].

The mechanism(s) of mtROS production responsible for 5-LOX activation remain unclear. We observed that the uncoupler of oxidative phosphorylation carbonyl cyanide 4-(trifluoromethoxy)phenylhydrazone (FCCP) and the respiration inhibitor antimycin A, which dissipate the mitochondrial membrane potential, strongly inhibit LTB4 synthesis induced by all tested stimuli [29]. These data suggest that voltage-dependent accumulation of Ca^2+^ in mitochondria may be important for the production of mtROS. Mitochondrial Ca^2+^ overload can induce the mPTP opening, but no effect of the mPTP inhibitor cyclosporine A on LTB4 synthesis was observed, indicating that the mPTP opening does not mediate mtROS production, which is responsible for 5-LOX stimulation, in contrast to stimulation of the oxidative burst and NETosis (see above). Multiple alternative mechanisms of Ca^2+^-dependent mtROS production have not been fully elucidated.

We have shown that the selective p38 inhibitor SB203580 inhibits fMLP-induced leukotriene synthesis, whereas the ERK1/2 activation inhibitor U0126 inhibits LTB4 synthesis induced by any of the tested stimuli. SkQ1 effectively prevents p38 and ERK1/2 activation (accumulation of phosphorylated forms), indicating that the activation of mitogen-activated protein kinases may be a critical target for mtROS involved in 5-LOXactivation [29].

Pharmacological inhibition of 5-LOX is highly effective in asthma [130] and beneficial in chronic inflammation [131]. Many 5-LOX inhibitors have been considered for these diseases, but only one, zileuton (trade name Zyflo), is widely used to treat asthma. Our data suggest that mitochondria-targeted antioxidants may be promising drugs for the treatment of pathologies associated with the dysregulation of leukotriene synthesis.

## 6. Apoptosis in Granulocytes

Neutrophils, eosinophils, and basophils are full of cyto- and histo-toxic substances and thus their elimination is properly regulated. Granulocytes are able to die by different mechanisms, which include apoptosis, necrosis, necroptosis, autophagy, pyroptosis, and ETosis [132]. Apoptosis is the only type of cell death that promotes the resolution of inflammation [133]. Under normal physiological conditions, neutrophil aging is accompanied by an increase in the expression of the chemokine receptor CXCR4 on the surface, which leads to the migration of neutrophils to bone marrow and liver, where they undergo apoptosis [134]. Spontaneous apoptosis of senescent neutrophils and their subsequent phagocytosis by macrophages (efferocytosis) prevent excessive inflammatory reactions [135].

Neutrophil apoptosis is tightly regulated in many ways, and mitochondria play a key role in cell fate decisions. The major mechanism of apoptosis (intrinsic pathway) depends on the release of cytochrome *c* and other apoptogenic factors from mitochondria into the cytoplasm [136]. Pro-apoptotic BCL-2 family proteins BAX and BAK mediate the permeabilization of the outer mitochondrial membrane, whereas anti-apoptotic members of the same family (MCL-1, A1/BFL-1, BCL-XL, BCL-2, and BCL-W) inhibit the release of mitochondrial apoptogenic factors and cell death [137]. So-called “BH3-only” protein members, which also belong to the BCL-2 family (BID, BIM, PUMA/BBC3, BAD, NOXA/PMAIP, BIK/BLK/NBK, BMF, and HRK/DP5) can induce apoptosis by competing with anti-apoptotic proteins or directly activating BAX/BAK [138].

Neutrophils have a trace amount of cytochrome *c*, but they are also much more susceptible to this protein, which promotes the assembly of apoptosome complex and subsequently activates effector caspase-3 [139]. The main anti-apoptotic protein in neutrophils is Mcl-1 [140]. The half-life of Mcl-1 is extremely short, whereas pro-apoptotic proteins (Bax, Bid, Bim) live longer [141], so in the absence of proinflammatory stimuli, a decrease in Mcl-1 level provokes apoptosis [142]. When neutrophils are exposed to proinflammatory factors (LPS, granulocyte-macrophage colony-stimulating factor (GM-CSF), and other cytokines), Mcl-1 expression is increased, whereas Bax and Bad are phosphorylated and inactivated, thus delaying apoptosis [143,144].

The regulation of neutrophil apoptosis by ROS still remains a controversial issue. It is known that hydrogen peroxide and hydroxyl radical accelerate spontaneous neutrophil apoptosis [145]. In line with this observation, the neutrophils from CGD patients with impaired ROS production have delayed apoptosis [146]. Phagocytosis of *Escherichia coli* in neutrophils accompanied by a NOX2-generated oxidative burst was reported to stimulate apoptosis [147], whereas spontaneous apoptosis does not depend on NOX2 activity [148]. NOX2-derived ROS can also stimulate the release of cathepsin D from azurophilic granules, followed by protolithic activation of caspase-8 and mitochondria-independent apoptosis [149]. On the other hand, there is evidence that the delay of apoptosis by many proinflammatory molecules (LPS, leukotriene B4, *N*-formylated proteins, IL-8) depends on ROS production by NOX2 [150]. LPS is known to activate the transcription factor NF-κB, which promotes neutrophil survival, and this activation relies upon ROS production [151]. The anti-apoptotic action of LTB4 is also mediated by NOX2-derived ROS and involves NF-kB activation [152].

The suppression of mtROS production by SkQ1 was found to promote the spontaneous apoptosis [153] and to accelerate the apoptosis of fMLP-stimulated neutrophils [51]. It was also shown that mtROS production in hypoxia stabilized transcription factor HIF-1α, which activated pro-survival NF-κB-dependent signaling so that mitochondria-targeted antioxidant Mito-TEMPO could accelerate neutrophil apoptosis [154]. Thus, mtROS can inhibit apoptosis. It is likely that the mtROS-dependent activation of NOX2 and degranulation described above is involved in the antiapoptotic effects of proinflammatory molecules.

The apoptosis of human eosinophils is regulated by mitochondria and by ROS [34]. It was shown that NOX2-derived ROS are responsible in spontaneous eosinophil apoptosis [155] and in apoptosis delayed by GM-CSF [156]. Contrary to these data, antioxidant *N*-acetyl-L-cysteine inhibited spontaneous eosinophil apoptosis but increased apoptosis in the presence of GM-CSF [157]. Mitochondrial damage and increased mtROS production were shown to accompany eosinophil apoptosis, and GM-CSF lowered mtROS production [158]. It was shown also that mitochondrial respiration is required for eosinophil apoptosis stimulated by hydrogen peroxide [159]. These data suggest that mtROS stimulate eosinophil apoptosis; however, additional evidence is necessary.

## 7. Conclusions

The comprehensive data presented in this review strongly support the central role of mitochondria in regulating various effector functions of granulocytes. Mitochondrial production of ROS is increased upon granulocyte stimulation, influencing critical signaling pathways that dictate granulocyte activity and lifespan. Dysregulation of granulocyte functions has been linked to various inflammatory, allergic, and autoimmune conditions. An important role in tumor progression is played by the infiltration and activation of neutrophils in the tumor environment. Neutrophils can stimulate tumor invasion and directly promote tumor growth [160]. Importantly, neutrophils have been shown to counteract antitumor immune responses by suppressing T cells and NK cells, mainly through the toxic effects of ROS [161,162]. NETs and NETosis also stimulate tumor growth [163,164]. It was reported by Rice et al. [165] that immature neutrophils with increased mitochondrial content and an active metabolism can preferentially infiltrate some tumors and survive in conditions of limited glucose supply to the tumor microenvironment. The authors suggested that mitochondria supply NADPH to stimulate NOX-dependent ROS production in these neutrophils and suppress the antitumor T cell immune response. The data discussed above suggest that mitochondrial ROS production may also be involved in the activation of NOX and NETosis, supporting tumor progression.

A wide variety of pharmacological interventions have been proposed to inhibit excessive granulocyte activation. Among these, the use of mitochondrial-targeting antioxidants shows promises as a therapeutic approach for pathologies where inflammation plays a significant role [166]. SkQ1, an efficient mitochondrial-targeting antioxidant, has demonstrated notable anti-inflammatory properties in models of acute bacterial infection [167] and systemic inflammatory response [168]. Furthermore, SkQ1 has exhibited therapeutic effects in preclinical models for cardiovascular and renal diseases [169,170]. Notably, SkQ1 eye drops have shown high efficacy in various models of inflammatory eye diseases [171,172], as well as in a clinical study addressing dry eye syndrome [173]. More recently, SkQ1 has been shown to prevent rapid death in mice induced by four very different shocks induced by LPS, intravenous mitochondrial injection, cooling, and toxic doses of C_12_TPP [174]. It is important to note that all of these stresses were accompanied by a strong increase in the level of pro-inflammatory cytokines and that SkQ1 prevented this cytokine storm. The results discussed here suggest that the ability of SkQ1 to inhibit effector functions and leukotriene production and stimulate granulocyte apoptosis contributes to its therapeutic effects. These exciting results pave the way for further research into the potential of mitochondria-targeted antioxidants for the treatment of pathologies associated with the excessive activation of granulocytes.

## Figures and Tables

**Figure 1 cells-12-02210-f001:**
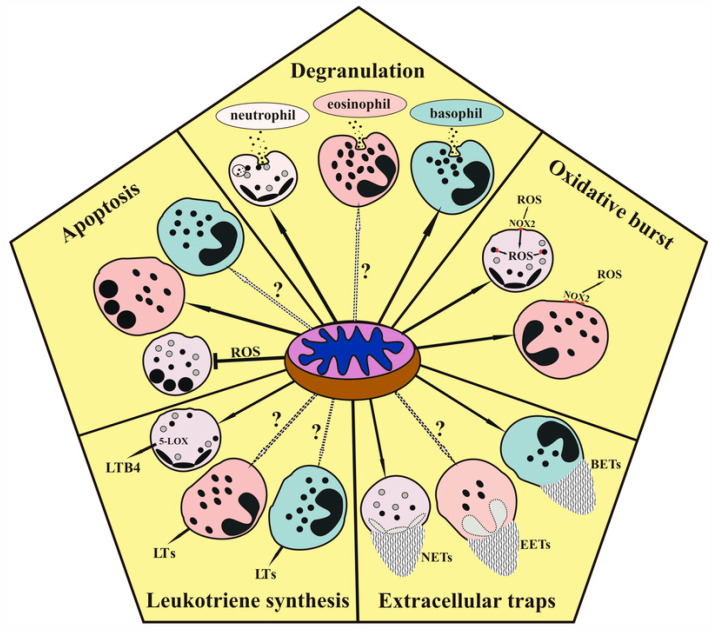
Scheme illustrating the mitochondrial control of granulocyte effector functions, leukotriene synthesis, and apoptosis. ROS, reactive oxygen species; NOX2, NADPH oxidase 2; NETs, EETs, BETs, extracellular DNA-containing traps released from neutrophils, eosinophils, and basophils, respectively; LTs, leukotrienes; LTB4, leukotriene B4. Interactions that have not been verified experimentally are marked with a “?”.

**Figure 2 cells-12-02210-f002:**
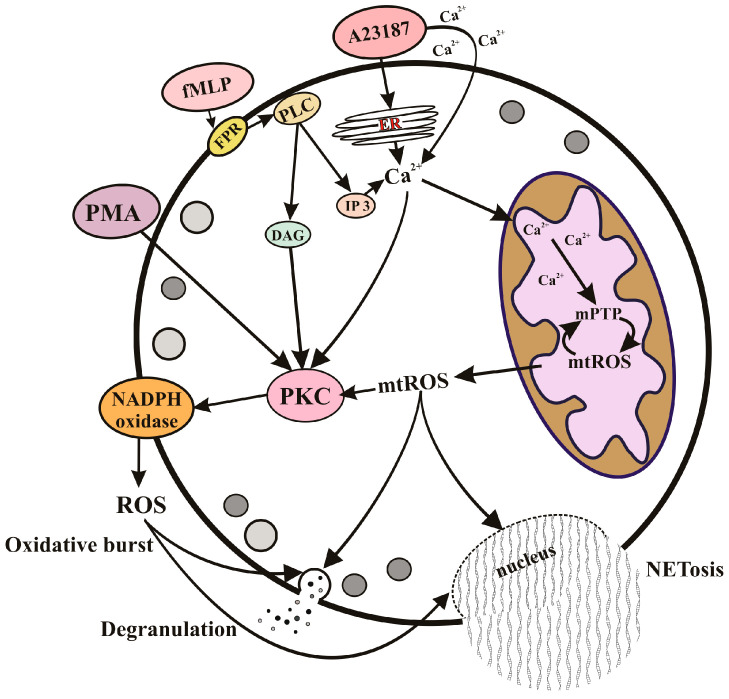
Regulation of effector functions of neutrophils by mitochondrial ROS. Degranulation, oxidative burst, and NETosis can be stimulated by pathogens through specific receptors (such as the specific G protein-coupled fMLP receptor) or by direct Ca^2+^ mobilization (as modeled with the calcium ionophore A23187). The subsequent accumulation of Ca^2+^ in the mitochondrial matrix leads to the opening of nonselective mitochondrial pores (mPTP) and the formation of mitochondrial ROS (mtROS). mtROS released into the cytosol can activate NADPH oxidase (NOX2), degranulation, and NETosis through protein kinase C (PKC) activation, as well as through several unidentified signaling pathways. ROS produced by NOX2 also stimulate degranulation and NETosis. Receptor-dependent activation includes stimulation of phospholipase C (PLC), which catalyzes the synthesis of inositol triphosphate (IP3) and diacylglycerol (DAG). DAG is a powerful stimulus for PKC, so this branch of neutrophil activation is independent of mitochondria. This is modeled by the synthetic analogue of DAG, phorbol 12-myristate 13-acetate (PMA), which stimulates the effector functions of neutrophils independently of mitochondria.

## Data Availability

Data sharing not applicable.

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
