# Peer review of "Role of Mitochondria in the Regulation of Effector Functions of Granulocytes"

_cells, 2023, doi:10.3390/cells12182210_

Round 1

Reviewer 1 Report

This review manuscript gives comprehensive overview of basic effector functions of granulocytes and summarizes the role of mitochondria ROS in regulating these effector functions by highlighting the research results from the recent publications. 

Minor comments:

The two sentences at the Lines 128-130 on Page 3 ("Possible role of mitochondria in production of ........ Our studies using SkQ1......LTB4 production in neutrophils.") may be removed because they are followed by the two very similar sentences which are better written and include the reference citation. 

This review manuscript gives comprehensive overview of basic effector functions of granulocytes and summarizes the role of mitochondria ROS in regulating these effector functions by highlighting the research results from the recent publications. 

Minor comments:

The two sentences at the Lines 128-130 on Page 3 ("Possible role of mitochondria in production of ........ Our studies using SkQ1......LTB4 production in neutrophils.") may be removed because they are followed by the two very similar sentences which are better written and include the reference citation. 

Author Response

The two sentences at the Lines 128-130 on Page 3 ("Possible role of mitochondria in production of ........ Our studies using SkQ1......LTB4 production in neutrophils.") may be removed because they are followed by the two very similar sentences which are better written and include the reference citation.

We are grateful to the reviewer for this comment. We removed two sentences that were repeated (lines 130-132).

Reviewer 2 Report

I have read the review by Vorobjeva et al. with great pleasure. The review reports the role of mitochondria in regulation of effector functions of granulocytes. Granulocytes were usually considered as cells with low content of mitochondria, and the role of these organells was underestimated. Recent knowledge about mitochondrial functions in neutrophils demonstrates their crucial contribution to neutrophil activation, degranulation, respiratory burst, NET formation and death. 

Several existing reviews highlight different aspects of this thema. One of the strong sides of the current manuscript in comparison to others, is the attention to eosinophils and basophils, usually underestimated in reviews dedicated to similar topics. On my opinion, data about eosinophils and basophils deserves to be separated into a chapter, to focus the attention on these cells. Another interesting aspect mentioned in the current manuscript, is regulation of leukotrienes synthesis by mitochondria, not overviewed by other authors. 

At the same time there are several aspects that still can be improved.

1. The coverage of knowledge about regulation of neutrophil functions by mitochondria is not complete. The important aspects include neutrophil development and differentiation, migration and adhesion, interaction with other cell types. 

2. Some of the statments in the conclusion are not fully discussed in the text.

"Dysregulation of granulocyte functions has been linked to various inflammatory, allergic, and autoimmune conditions..." (567-568) - the diseases associated with mitochondrial dysregulation and thus impaired neutrophilic functions are scarcely mentioned. One of the important aspects nowadays is cancer-associated mitochondrial changes and neutrophil dysfunction (eg https://doi.org/10.1038/s41467-018-07505-2).

"A wide variety of pharmacological interventions have been proposed..." (568-569) - it would be helpful to have an overview of inhibitors summarized in one chapter.

3. The excessive focus at details of the functionality, with relatively fewer attention at mitochondria which should be the main scope of the review. In some cases, after some detailed information (eg types of exocytosis, mechanism of degranulation) there is no following discussion how mitochondria regulate thoose particular processes. 

4. The manuscript requires more clear structure. Eg in introduction there are facts about mitochondria, repeated in the following chapters. 

5. The manuscript requires figures or tables summarizing the major facts, what will make the review easier understandable.

6. Low percent of citiations of recently (2018-2023) published manuscripts on the topic.

Author Response

On my opinion, data about eosinophils and basophils deserves to be separated into a chapter, to focus the attention on these cells.

We are grateful to the reviewer for the high evaluation of our manuscript. We have not dedicated a separate chapter for data on eosinophils and basophils for two reasons: (1) this would inevitably lead to repetition in the description of effector functions; (2) not so much is known about the role of mitochondria in regulation of eosinophils and basophils functions.

At the same time there are several aspects that still can be improved.

  1. The coverage of knowledge about regulation of neutrophil functions by mitochondria is not complete. The important aspects include neutrophil development and differentiation, migration and adhesion, interaction with other cell types. 

We agree with the reviewer that differentiation, migration, and adhesion of granulocytes, as well as their interaction with other cell types, deserve special analysis. However, there are practically no experimental data on the role of mitochondria in these processes. Therefore, we limited ourselves to considering only the effector functions of granulocytes and their apoptosis.

  1. Some of the statements in the conclusion are not fully discussed in the text.

"Dysregulation of granulocyte functions has been linked to various inflammatory, allergic, and autoimmune conditions..." (567-568) - the diseases associated with mitochondrial dysregulation and thus impaired neutrophilic functions are scarcely mentioned. One of the important aspects nowadays is cancer-associated mitochondrial changes and neutrophil dysfunction (eg https://doi.org/10.1038/s41467-018-07505-2).

We are grateful to the reviewer for this important suggestion. We are aware of studies on tumor-associated neutrophils, but we missed the work indicated by the reviewer. To the best of our knowledge, this is the only report indicating a possible role for mitochondria in granulocytes to be associated with tumors and promote tumor growth. We have added the next paragraph as well as new references to the Conclusion section.

An important role in tumor progression is played by infiltration and activation of neu-trophils in the tumor environment. Neutrophils can stimulate tumor invasion and di-rectly promote tumor growth [160]. Importantly, neutrophils have been shown to counteract antitumor immune responses by suppressing T cells and NK cells, mainly through the toxic effects of ROS [161,162]. NETs and NETosis also stimulate tumor growth [163,164]. It was reported by Rice et al. [165], that immature neutrophils with increased mitochondrial content and active metabolism can preferentially infiltrate some tumors and survive in conditions of limited glucose supply to the tumor micro-environment. The authors suggested that mitochondria supply NADPH to stimulate NOX-dependent ROS production in these neutrophils and suppress the antitumor T-cell immune response. The data discussed above suggest that mitochondrial ROS produc-tion may also be involved in the activation of NOX and NETosis, supporting tumor progression.

  1. Groth, C.; Weber, R.; Lasser, S.; Özbay, F.G.; Kurzay, A.; Petrova, V.; Altevogt, P.; Utikal, J.; Umansky, V. Tumor Promoting Capacity of Polymorphonuclear Myeloid-Derived Suppressor Cells and Their Neutralization. J. Cancer 2021, 149, 1628–1638, doi:10.1002/ijc.33731.
  2. Schmielau, J.; Finn, O.J. Activated Granulocytes and Granulocyte-Derived Hydrogen Peroxide Are the Underlying Mechanism of Suppression of T-Cell Function in Advanced Cancer Patients. Cancer Res. 2001, 61, 4756–4760.
  3. Bhardwaj, V.; Ansell, S.M. Modulation of T-Cell Function by Myeloid-Derived Suppressor Cells in Hematological Malignancies. Cell Dev. Biol. 2023, 11, 1129343, doi:10.3389/fcell.2023.1129343.
  4. Yazdani, H.O.; Roy, E.; Comerci, A.J.; van der Windt, D.J.; Zhang, H.; Huang, H.; Loughran, P.; Shiva, S.; Geller, D.A.; Bartlett, D.L.; et al. Neutrophil Extracellular Traps Drive Mitochondrial Homeostasis in Tumors to Augment Growth. Cancer Res. 2019, 79, 5626–5639, doi:10.1158/0008-5472.CAN-19-0800.
  5. Cristinziano, L.; Modestino, L.; Antonelli, A.; Marone, G.; Simon, H.-U.; Varricchi, G.; Galdiero, M.R. Neutrophil Extracellular Traps in Cancer. Cancer Biol. 2022, 79, 91–104, doi:10.1016/j.semcancer.2021.07.011.
  6. Rice, C.M.; Davies, L.C.; Subleski, J.J.; Maio, N.; Gonzalez-Cotto, M.; Andrews, C.; Patel, N.L.; Palmieri, E.M.; Weiss, J.M.; Lee, J.-M.; et al. Tumour-Elicited Neutrophils Engage Mitochondrial Metabolism to Circumvent Nutrient Limitations and Maintain Immune Suppression. Commun. 2018, 9, 5099, doi:10.1038/s41467-018-07505-2.

"A wide variety of pharmacological interventions have been proposed..." (568-569) - it would be helpful to have an overview of inhibitors summarized in one chapter.

We are grateful to the reviewer for this suggestion. Indeed, many pharmacological inhibitors of granulocyte function are known and widely used for the treatment of inflammatory diseases. However, to our knowledge, no drugs have been proposed that target mitochondria to suppress granulocytes. Research using mitochondria-targeted antioxidants suggests that this line of research could be very promising. These studies are discussed in detail in our review. We believe that it would be redundant to cover here the vast area of antigranulocytic pharmacology.

  1. The excessive focus at details of the functionality, with relatively fewer attention at mitochondria which should be the main scope of the review. In some cases, after some detailed information (eg types of exocytosis, mechanism of degranulation) there is no following discussion how mitochondria regulate thoose particular processes.

We fully agree with this reviewer's comment. Unfortunately, this shortcoming of our review reflects a general gap in knowledge about the mechanisms of mitochondrial control of various functions in granulocytes, as well as in many other cell types. Signaling pathways and specific targets modulated by mitochondrial ROS and metabolites have not yet been identified. The most pronounced progress has been made in recent studies on macrophages, but these results cannot be directly applied to other cells.

  1. The manuscript requires more clear structure. Eg in introduction there are facts about mitochondria, repeated in the following chapters.

We are grateful to the reviewer for this suggestion.  We removed some repeats (lines 130-132) and added the small fragment marked in green.

  1. The manuscript requires figures or tables summarizing the major facts, what will make the review easier understandable.

We fully agree with the reviewer and editor. Two schemes have been added to the revised version.

Figure 1. Scheme illustrating the mitochondrial control of granulocyte effector functions, leukotriene synthesis, and apoptosis.

Figure 2. Regulation of effector functions of neutrophils by mitochondrial ROS.

  1. Low percent of citiations of recently (2018-2023) published manuscripts on the topic.

We agree with this reviewer's comment. However, the bibliography reflects the state of the art in this field. To the best of our knowledge, no significant progress has been made in studying the mitochondrial regulation of granulocyte functions. This is partly a result of the impossibility of genetic manipulation in these cells. Unfortunately, the use of specific pharmacological agents targeted to mitochondria is very limited. Almost all studies in this direction are cited in our review.

Reviewer 3 Report

The authors summarized the role of mitochondria and mitochondria-derived reactive oxygen species (ROS) in regulating granulocyte functional responses. The review is well-written and organized. I don't have any comments. I congratulate the authors on this informative review. 

Author Response

We thank the reviewer for the high evaluation of our manuscript.

Round 2

Reviewer 2 Report

Please add description of the abbreviatures used in the Fig.1, in the figure legend.

Author Response

We are grateful to your comments. We have added the description of the abbreviatures used in the Fig.1, in the figure legend.

Fig. 1.  Scheme illustrating the mitochondrial control of granulocyte effector functions, leukotriene synthesis, and apoptosis.

ROS, reactive oxygen species; NOX2, NADPH oxidase 2; NETs, EETs, BETs, extracellular DNA-containing traps released from neutrophils, eosinophils and basophils, respectively; LTs, leukotrienes; LTB4, leukotriene B4.